# NORMDIAL: A Comparable Bilingual Synthetic Dialogue Dataset for Modeling Social Norm Adherence and Violation

**Oliver Li**[°]   **Mallika Subramanian**[°]
**Arkadiy Saakyan**[°]   **Sky CH-Wang**[°]   **Smaranda Muresan**[°][•]
[°]Department of Computer Science, Columbia University
[•]Data Science Institute, Columbia University
{al4143, ms6544, smara}@columbia.edu, {a.saakyan, skywang}@cs.columbia.edu

## Abstract

Social norms fundamentally shape interpersonal communication. We present NORMDIAL, a high-quality dyadic dialogue dataset with turn-by-turn annotations of social norm adherences and violations for Chinese and American cultures. Introducing the task of social norm observance detection, our dataset is synthetically generated in both Chinese and English using a human-in-the-loop pipeline by prompting large language models with a small collection of expert-annotated social norms. We show that our generated dialogues are of high quality through human evaluation and further evaluate the performance of existing large language models on this task. Our findings point towards new directions for understanding the nuances of social norms as they manifest in conversational contexts that span across languages and cultures.

## 1 Introduction

Social norms—implicitly learned notions of acceptable behavior—both develop from and guide our everyday interactions (Sherif, 1936). As with the value systems that underlie these notions, the acceptability and deemed typicality of behaviors varies across cultures (Triandis et al., 1994). For example, due to a strong emphasis on individualism, open and direct expression of opinions and disagreement is often encouraged and valued in Western cultures (Arieli, 1964), while such acts may often be viewed as disruptive to social order in Eastern Asian cultures that value collectivism (Triandis, 1993). Understanding these cultural nuances is key to empower computational systems to reason across cultural contexts (Liu et al., 2021).

We introduce NORMDIAL, a bilingual synthetically generated dyadic dialogue dataset of social norms as they appear within the context of conversational interactions. Gathering realistic data at scale in this domain presents a challenging and potentially cost-prohibitive task, particularly in the

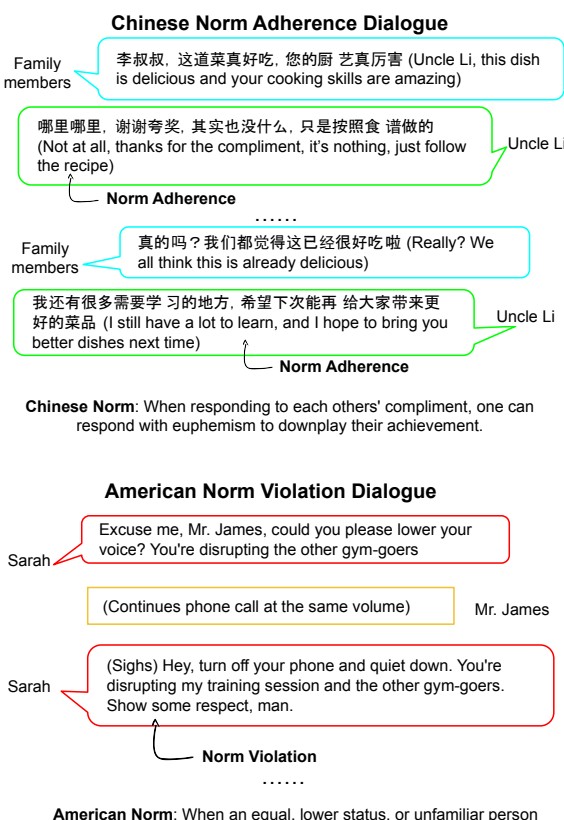

Figure 1: Examples of generated dialogues with adherences (top, in Chinese) and violations (bottom, in English) to social norms about responding to compliments and giving requests, respectively.

context of identifying social norm adherences and violations across multiple cultural contexts. This paucity of data hinders progress towards developing cross-cultural communication tools.

As a small step towards addressing this gap, leveraging recent successes in utilizing large language models (LLMs) for social data generation and augmentation (Kim et al., 2022a; Chen et al., 2023), we propose a human-in-the-loop framework to synthesize realistic conversational data under expert prompting for modeling social norm adher-

ence and violation. Using this human-AI collaboration framework, we generate a series of 4231 dyadic dialogues totaling 29550 conversational turns grounded in theoretical norm categories (Linguistic Data Consortium, 2022) across Chinese and American cultures, and demonstrate that our synthetic bilingual conversations are comparable to or exceed the quality of existing, naturally occurring datasets under interactive human evaluation and automatic metrics; examples of our dialogues are shown in Figure 1. Our dataset presents social norm adherences and violations labeled on a dialogue-turn basis; with this labeling task decoupled from dialogue generation, we further evaluate the capability of existing LLMs in reasoning about norm adherences and violations in a conversational setting and show that existing models often fail to reason correctly about these contexts. We hope that this resource will further motivate research towards designing better systems able to promote more fluid cross-cultural conversational interactions. We make NormDial available at https://github.com/Aochong-Li/NormDial.

## 2 Background and Related Work

**LLMs for Synthetic Data Generation.** Prompting LLMs to synthesize and augment language data for existing tasks (Li et al., 2022; Møller et al., 2023; Chen et al., 2023) has emerged as a viable, cost-effective alternative in lieu of crowd-sourced annotation at scale or alternative strategies such as fine-tuning language generators (Papangelis et al., 2021; Zhang et al., 2020) in the dialogue domain. LLMs, trained on massive amounts of web text, suffer from representational and allocational harms (Blodgett et al., 2020; Weidinger et al., 2021). Yet, such models often also possess high algorithmic fidelity in the realm of representing latent social variables (Argyle et al., 2023), in that these sources of bias may often be finely controlled for to accurately emulate responses from a variety of human demographic sub-populations in areas such as predicting historically missing survey responses in social research (Kim and Lee, 2023). Here, under this vein, we employ a human-in-the-loop framework to both finely condition and validate the generation of dialogues for modeling social norms.

**Computational Social Norms.** Our work is situated in the broader push towards empowering computational systems of interaction with the capability to reason in socio-culturally situated contexts

(Ziems et al., 2023), spanning commonsense reasoning (Sap et al., 2019; Rashkin et al., 2018), the determination of appropriate and morally ethical behavior (Emelin et al., 2021; Jiang et al., 2022), and the further grounding of this behavior in areas like dialogue systems and situated question answering (Kim et al., 2022b; Ziems et al., 2022; Gu et al., 2022) more specifically on underlying knowledge of social norms. While most work on computational models of social norms has been focused on the American context (Forbes et al., 2020), recent work has begun to bridge this gap cross-culturally to enable a comparison of descriptive nuances in norms across cultures (CH-Wang et al., 2023; Fung et al., 2022). Here, our work builds a dialog dataset around conversational social norms for both American and Chinese cultures.

## 3 The NORMDIAL Dataset

NORMDIAL is a human-in-the-loop synthetically generated bilingual (Chinese & English) dyadic dialogue dataset for studying social norms as they appear in different conversational contexts. Dialogue turns are further labeled on whether they adhere to or violate a given social norm with textual explanations. Our human-AI collaboration framework for creating NORMDIAL is shown in Figure 2.

**Social Norm Augmentation (Stage 0 in Figure 2).** The Linguistic Data Consortium (LDC) (Linguistic Data Consortium, 2022) taxonomizes 10 categorizations of social norms in its guidelines—apology, compliment, condolence, criticism, greeting, leave, persuasion, request, response to compliment, giving thanks—and provides a detailed set of associated norms (5 for each category) for Chinese culture. Examples of verbal evidence of adherence to a norm in a conversational context, alongside the relationship data of each hypothetical interlocutor, are provided as details for norm definitions.

We augment this starting set of validated social norms by in-context prompting ChatGPT (Wei et al., 2022), making use of LDC norm descriptions and examples in our prompt as few-shot in-context examples, to generate a greater set of social norms for Chinese culture, which are then conditioned on and further prompted to generate corresponding norms for American culture. To verify the correctness and applicability of generated norms for each cultural context, we task annotators who identify as native speakers of each respective language and who have significant (10+ years) lived experiences

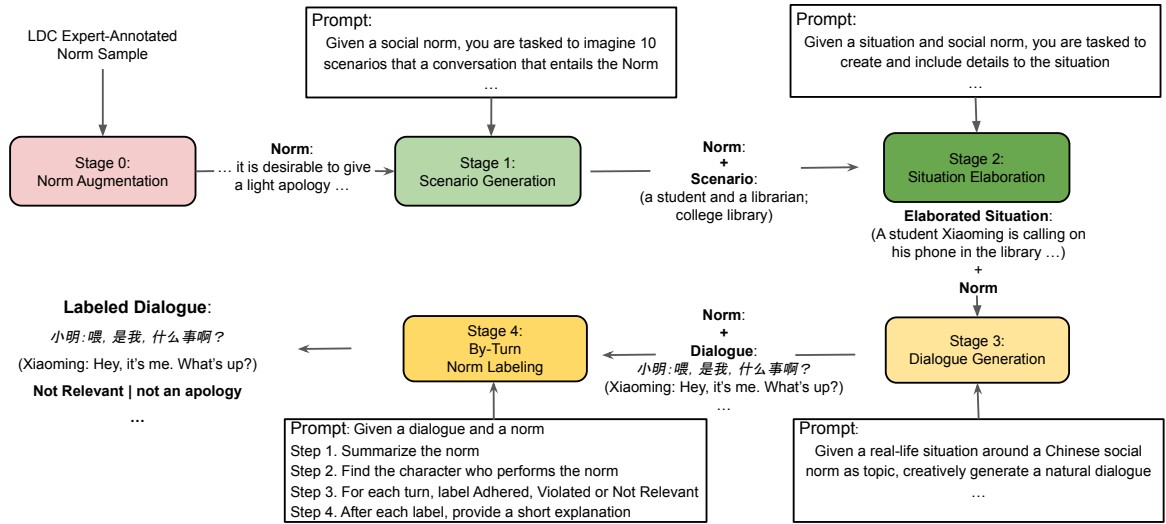

Figure 2: Our human-AI collaboration framework for creating NORMDIAL, through (0) norm augmentation with a small set of expert-labeled social norms under the LDC taxonomy, (1) scenario generation, (2) situation elaboration, (3) dialogue generation, and (4) turn-by-turn norm labeling with human verification at every stage. We show a Chinese dialogue generation example here for illustration; the same pipeline is adopted for English dialogues.

in each culture to manually evaluate and rectify if each generated norm is (1) factually correct according to their own lived experiences, (2) in line with the defined norm category, (3) specific to the culture, and (4) detailed in its description, removing those that did not satisfy these criteria. This process enables us to collect a dataset of 133 and 134 Chinese and American social norms, respectively (see Table 1). Additional norm examples alongside the prompts used are shown in Appendix B.

**Chinese Norm (Respond to Compliments)** : When a person of lower status respond to the compliments of one of high status, one can say "没有我还有很多不足,以后多向前辈请教和学习" (I still have many short-comings, I'll seek advice and learn from my seniors.)

**American Norm (Respond to Compliments)**: When a person of lower status responds to a compliment from someone of higher status, it is common to express gratitude and acknowledge the compliment gracefully. [...]

Table 1: Examples of manually verified Chinese and American norms generated by ChatGPT.

**Scenario Imagination and Situation Elaboration (Stages 1 and 2 in Figure 2)** Social norms manifest in different ways depending on the conversational context (Lockshin and Williams, 2020). An issue in dialogue generation from a small amount of hand-written data is its lack of diversity, as in-context examples have a large impact on prompting results (Min et al., 2022). To tackle this issue, with our augmented set of social norms, we first prompt

ChatGPT using one-shot learning to generate a list of 10 short scenarios in the form of *social relation; location* where given norms are most likely to take place in real life. In the second stage, we combine each scenario with a given social norm to enable ChatGPT to elaborate on and expand each scenario description into ones that are more detailed and realistic. In total, we obtained 4231 unique situations from this process; the topics of elaborated situations as captured by a 30-topic Latent Dirichlet Allocation (LDA) model are presented in Appendix E. To ensure that situations are elaborated faithfully from the given norm, we collected a sample of 218 situations along with their norms for three annotators to verify if each situation entails the norm. The results in Appendix D show high faithfulness scores, with the lowest for American norm violations. For the final version of NORMDIAL, we manually verify and remove situations that deviate from the norm descriptions (releasing both raw and cleaned datasets).

**Dialogue Generation (Stage 3 in Figure 2).** By prompting ChatGPT with pairs of norms and their elaborated situations along with an in-context example, we generate turn-by-turn dyadic dialogues that either adhere to or violate the given social norm. Shown in Figure 3, we find that CoT prompting with Scenario Generation (Stage 1) and Situation Elaboration (Stage 2) greatly improves dialogue lexical diversity as compared to directly

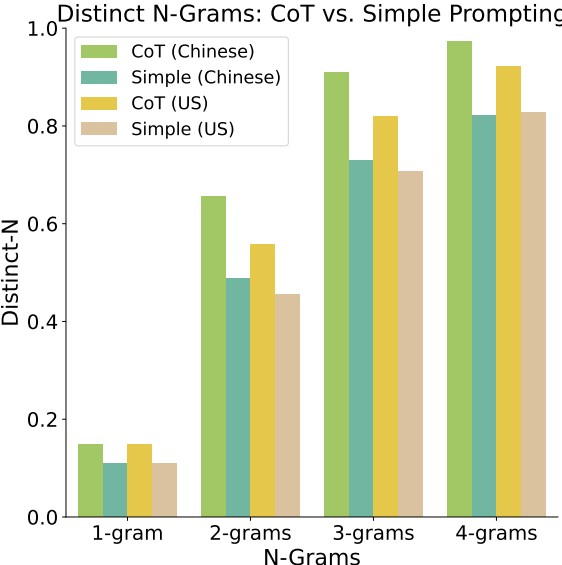

Figure 3: Distinct N-grams for both US and Chinese generated dialogues under Simple and CoT prompting. CoT prompting with situation elaboration improves dialogue diversity compared to Simple prompting without using situations. Chinese texts are tokenized by `jieba`.[1]

generating dialogues from norms alone (Simple), measured by distinct N-grams. Prompts used at each stage are provided in Appendix C.

**Automatic Turn-level Annotation of Norm Adherence and Violation (Stage 4 in Figure 2).** With this set of situationally grounded dyadic dialogues, we then further prompt ChatGPT using Chain-of-Thought (CoT) (Wei et al., 2022) reasoning to label whether each dialogue turn (1) adheres to, (2) violates, or (3) is not relevant to the given norm. Providing the social norm, situation, and dialogue, we prompt ChatGPT to (1) summarize the norm into a short description of its rules and actions as the *norm action*; (2) respond with the names of the characters who are mostly involved with acting in alignment with the norm as the *norm actors*; and (3), for each dialogue turn, with information about the *norm action* and *norm actors*, predict the label (*Adhered*, *Violated*, or *Not Relevant*) together with a short textual explanation of each decision.

In the next Section, we discuss the evaluation of generated dialogue quality and automatic turn-level annotations of adherences and violations.

## 4 Evaluation

We conduct a human evaluation on the quality of dialogues and the correctness of model-predicted dialogue-turn norm adherence and violation labels with a pool of 12 annotators, each possessing significant lived experiences in both cultures.

**Synthetic Dialogue Quality.** To determine the quality of our Chinese and American synthetic dialogues, we perform a human evaluation against two types of baselines: (1) specifically curated domain-relevant dialogues or pre-existing and naturally occurring dialogue datasets, and (2) human-written dialogues specifically for our task.

For the former, we compare our Chinese dialogues against a collection of 200 dialogues randomly sampled from the Linguistic Data Consortium (LDC), which contains 413 Chinese conversations with expert-annotated turn-by-turn Chinese norm observations from video transcripts. These conversations contain dialogues and chitchat from sources such as TV shows, Vlogs, and other types of videos from Bilibili, a Chinese video-sharing platform. For English dialogues, as no existing domain-comparable dialogue is available, we compare against a set of 200 realistic, human-written dialogues reflecting daily communication that covers various topics about daily life, DailyDialog (Li et al., 2017); a comparison that has been used for previous evaluations of synthetic dialogue quality (Chen et al., 2023).

For the latter, to evaluate our dialogues against human-written counterparts specific to our task, we asked three native speakers of Chinese and English to creatively write a set of 20 dialogues for each language, based on a sample of 20 norms and situations, which we selected from each of our 10 norm categories by randomly selecting an adherence and violation (*norm, situation*) pair from each category.

We ask sets of 3 annotators to rate each conversation, formatted consistently, on their (1) naturalness, or how natural each dialogue sounded, (2) nativeness, if they thought the dialogue came from a native speaker, (3) coherence, and (4) interestingness, each on a 5-point Likert scale, with final scores for each aspect being the average of the scores received. On a separate task, we ask annotators to rate if synthetic dialogues were faithful and on-topic to their provided social norm, i.e. *does the main topic of the dialogue match the provided social norm, yes/no?*, of which final labels are obtained via majority-voting. A comparison of quality evaluation scores is shown in Table 2 and details about evaluation metrics are shown in Appendix Section A.

---

[1] https://github.com/fxsjy/jieba

| Sources | Natural | Native | Coherent | Interesting |
|---|---|---|---|---|
| Ours (ZH) | 3.78 | 3.78 | 3.94 | 3.66 |
| LDC | 3.69 | 3.81 | 3.28 | 2.53 |
| Human (ZH) | **4.50** | **4.60** | **4.55** | **4.25** |
| Ours (EN) | **4.41** | 4.78 | **4.85** | 4.22 |
| DailyDialog | 4.39 | 4.40 | 4.72 | **4.34** |
| Human (EN) | 4.15 | **4.95** | 4.20 | 3.95 |

Table 2: Dialogue quality evaluation across NORMDIAL synthetic dialogues (Ours), established and domain-specific baselines (LDC and DailyDialog), and human-written baselines (Human). Chinese language data is marked as ZH; English as EN.

For baseline (1), Annotators rate NORMDIAL dialogues higher in almost all dialogue quality aspects than their pre-existing curated and naturally occurring dialogue baseline counterparts, with only DailyDialog outperforming NORMDIAL in interestingness and LDC outperforming our synthetic dialogues in nativeness; synthetic dialogues were rated higher to a statistically significant degree in coherence and interestingness for Chinese and nativeness and coherence for the American side. As for benchmark (2), NORMDIAL dialogues were found to be of higher quality than their specifically tasked human written counterparts for English and lower for Chinese, in line with language performance differences for ChatGPT. Despite this performance difference in Chinese dialogues, it took the average annotator more than an hour to finish writing 20 dialogues; as this is a highly creative task, tasking annotators in our task can prove to be a challenge in scalability, given emerging evidence of annotator fatigue (Derczynski et al., 2020), especially for creative tasks.[2] On the other hand, taking the majority vote of dialogue on-topic labels from annotators showed that 92.5% and 86.5% of dialogues for Chinese and English, respectively, were faithful to their prompted (norm, situation) pairs.

**Automatic Turn-level Annotation.** As our automatic dialogue-turn norm adherence/violation labeling via prompting is separate from dialogue generation, a natural question arises as to how well existing LLMs are able to perform this task, i.e., *how well can LLMs accurately detect if a given social norm is adhered to or violated for a conversational round*? Here, collecting a sample of 200 dialogues for each language, two annotators manually validated the correctness of ChatGPT labeled dialogue rounds to check for correctness, resolv-

ing disagreements via discussion to produce a set of final gold standard labels for 1281 Chinese and 1503 English dialogue rounds. Table 3 shows the precision, recall, and F1 scores of ChatGPT predictions against ground truth labels, stratified across dialogue language and label categories.

| Chinese | | | |
|---|---|---|---|
| Norm Labels | Precision | Recall | F1-Score |
| Adhered | 78.4% | 84.3% | 0.81 |
| Not Relevant | 94.4% | 80.7% | 0.87 |
| Violated | 53.6% | 85.6% | 0.66 |
| English | | | |
| Adhered | 77.0% | 89.7% | 0.83 |
| Not Relevant | 95.9% | 68.6% | 0.80 |
| Violated | 51.2% | 98.8% | 0.68 |

Table 3: ChatGPT norm adherence and violation label prediction performance against annotator-corrected gold labels.

Shown in Table 3, empirically, ChatGPT achieved a higher F1 score on correctly predicting if a dialogue round adhered to or was not relevant to a given norm, but performed significantly worse in predicting norm violations for both languages. Given *violation*'s high recall, conversational turns that are not violations of the norm were falsely labeled as so, even with few-shot expert prompting. Under qualitative examination, we found that many of the turns that were falsely labeled as violations served to instead provide *context* before the actual violations rather than the violation behavior itself, suggesting the potential for further future improvement in this area.

## 5 Conclusion

We presented NORMDIAL, a synthetic, validated, high-quality dyadic dialogue dataset with turn-by-turn annotations of social norm adherences and violations for Chinese and American cultures in Chinese and English. Our evaluation of synthetic dialogue quality reveals that our dataset is comparable to and/or exceeds the quality of naturally occurring and domain-specific dialogue datasets. Furthermore, our analysis of LLM predictions of norm observance reveals areas for existing models to improve in this domain. Our resource points towards new directions for understanding the nuances of social norms as they manifest in conversational contexts that span across languages and cultures.

---

[2]https://www.reddit.com/r/ProlificAc/comments/17btpjs/

## Limitations

**Annotation Bias.** While we have augmented our synthetic data generation pipeline with human validation at every stage from individuals who possess significant lived experiences in Chinese and American cultural contexts to ensure correctness, it is important to acknowledge that our ultimate *representation* of the views and values of these cultures is limited *to* these lived experiences. In working towards more culturally representative studies, it is important to broaden to the views and values of those who are represented in experimental data and acknowledge the presence of further *intra*-cultural variations (Plank, 2022).

**Language Model Bias.** As with that which has been aforementioned, language models also possess sources of bias arising from the fundamental trained behavior of the tendency to mimic patterns in their training data. As such, it is important to critically question and challenge the viewpoints of those who are represented and reproduced within and which may seep into our dataset as a result, even under significant human validation.

## Ethical Considerations

**Names as Sources of Bias.** Within our human evaluation and annotation, a deliberate measure was implemented to address the potential introduction of biases by excluding character names during dialogue rounds. The purpose of this approach was to minimize the potential impact of personal biases or preconceived notions that may arise from specific names, ethnic backgrounds, or genders. As a result, annotators were solely guided by the dialogue's content and the cultural norms under discussion. In our data release, we emphasize the same need for future work to undertake similar measures to mitigate such sources of bias in annotation.

**Social Norms.** Cultural nuances are complex and multifaceted and do not remain static across time. Here, we have collaborated with social science researchers with significant established expertise in Chinese and American cultures to ensure the accuracy and validity of our set of social norms under rigorous verification, further consulting and adapt to the taxonomy provided by the LDC in defining and characterizing social norms. It is important to note here that it is impossible to have a "ground truth" set of social norms for every culture, as they are by nature aggregated judgments of acceptability that are subject to variation across longitudinal scales. Nonetheless, a central contribution of this work is a framework to create faithful dialogues that are themselves based on any given set of social norms, which allows for a "plug-and-play" dialogue generation for any additional/given set of social norms.

## Acknowledgements

We thank Maximillian Chen and Bo Feng for their helpful comments, thoughts, and discussions; Winston Wu, Zoie Zhao, Shiyu Zhang, Gechen Shen, Anxin Yi, Feiyang Zhu, Christopher Lee, Aniv Ray, Batool Taraif, and other anonymous crowdworkers for their help in annotations and evaluation; and the anonymous reviewers for their helpful feedback. This research is being developed with funding from the Defense Advanced Research Projects Agency (DARPA) CCU Program No. HR001122C0034. The views, opinions and/or findings expressed are those of the authors and should not be interpreted as representing the official views or policies of the Department of Defense or the U.S. Government. Sky CH-Wang is supported by a National Science Foundation Graduate Research Fellowship under Grant No. DGE-2036197.

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

## A  Dialogue Quality Evaluation Metrics

Annotators were asked to rate conversations from both synthetic sources and actual dialogue sources according to the following dimensions and instructions;

- **On-Topic**: *Does the dialogue demonstrate the stated social norm?*
  Option: 1 (Yes) or 0 (No)

- **Naturalness**: *How natural is the overall dialogue? How realistic does the content sound?*
  Scale: 1 (completely unnatural) to 5 (as natural as conversations in real life)

- **Nativeness**: *Does the dialogue sound native in its language? (If it uses any idioms, for instance, it might sound more native.)*
  Scale: 1 (not native at all) to 5 (as native as what a native Chinese speaker would say)

- **Coherent**: *How coherent is the overall dialogue? (e.g., If it contains illogical responses, it might not be coherent.)*
  Scale: 1 (completely incoherent) to 5 (as coherent as a native speaker)

- **Interesting**: *How interesting is the overall dialogue? Is the dialogue full of content?*
  Scale: 1 (generic and dull) to 5 (full of content and very engaging)

Each dialogue is evaluated by 3 crowd workers. Final scores for On-Topic are determined by the majority vote of all scores; for the other four dimensions, final scores are the unweighted average of all three annotator ratings.

## B  Social Norm Generation Prompts

### B.1  Prompts for Generating Chinese Norms

Social norms are informal rules that govern behaviors in groups and societies. Different conversational social norms are applicable to different conversation types. Imagine you are a culture-aware system that understands social norms in Chinese society. Using some examples provided, you are tasked to list down and describe 10 new conversational social norms that are related and specific to the conversation type given. While generating additional norms keep the following 3 instructions in mind:

1. Ensure that the norms generated are specific to the Chinese culture and are not generic moral or

conversational norms.

2. Specify the context where the norm is to be followed, wherever required.

3. Mention verbal pieces of evidence in Chinese that should be used in conversation for the accompanying norms.

## B.2 Prompts for Generating American Norms

Social norms are informal rules that govern behaviors in groups and societies. Different conversational social norms are applicable to different conversation types. Imagine you are a culture-aware system that understands social norms in American society. You are tasked to check whether a given set of norms for the Chinese culture are aligned to the American culture as well, or if they differ.

For each of the Chinese norms, if there exists an aligned social norm in American culture, generate the equivalent norm. If the American norm differs from the Chinese norm, then, generate the difference in the norm. In the norm descriptions generated, also include verbal phrases of evidence from an American culture that support the norm, if any. Do not list these down separately, include them in the norm description itself.

Conversation Type: <Norm Category>

Chinese Culture Norm: <Chinese Norm>

American Culture Norm:

## C Dialog Generation Pipeline Prompts

### C.1 Prompt for Scenario Generation

Social norms are informal rules that govern behaviors in groups and societies. You are given a social norm, and you are tasked to imagine 10 scenarios that a conversation that entails the Norm can take place in a real-life setting of Chinese society.

Format: Start your response with "Scenario:"

Norm: It is socially preferred to apologize immediately if you disturb another person and give the affected person a chance to identify and specify if they are hurt

Scenario: 1. in a university; college students 2. on the street; strangers 3. in a company's office; colleagues 4. in a hospital; patient and doctors 5. in a restaurant; waiter and customers 6. in a cafe; two customers 7. in a shopping mall; sales associates and customers 8. in a park; a morning jogger and a lady 9. in a suburb neighborhood; two neighbors who know each other 10. in a family gathering; two cousins

## C.2 Prompt for Situation Expansion

Social norms are informal rules that govern behaviors in groups and societies. You are given a situation and social norm, and you are tasked to create and include details to the real-life situation which takes place in Chinese society.

Format: start with "New Situation."

Norm: It is socially preferred to apologize immediately if you disturb another person and give the affected person a chance to identify and specify if they are hurt Situation: On the street; a Chinese young man and a woman

New Situation: A Chinese young man, 大伟, on his way back home, bumped into a stranger named Susan on the street. Susan is from New York, America, and it is her first time coming to China looking for her friend, so she doesn't speak fluent Chinese and is lost on the street.

## C.3 Prompt for Chinese Dialogue Generation

每次请根据一个围绕着中国社会规范的生活情景，有创意地生成一段人物间真实自然的对话脚本。

要求:

1. 对话中提及情景中所有细节和内容

2. 只需要生成对话脚本不需要额外解释

3. 且请以"对话"为开头生成对话，以"[结束]"标注对话结尾。

规范：如果你妨碍到了另一个人，你应该道歉并且询问对方以表示关心

情境：中国新年期间，一个中国小伙子大伟在王府井街上不小心撞到了纽约来找朋友的女人苏珊，大伟多次询问了苏珊是否受伤表示关心并且多次道歉。苏珊也同样地询问大伟是否被妨碍到，并且大伟因看到苏珊作为美国人说中文表示很新奇。

对话

大伟和苏珊: 哎呀

大伟: 哎呦，对不起，没撞到您吧

苏珊: 没事没事，真对不起

大伟: 没想到您还说中国话呢，您好

苏珊: 你好

大伟: 我刚才没碰到你吧?

苏珊: 我很好，就是不会走路，你还好吗

大伟: 我没事，新年快乐，注意安全

### C.4 Prompt for Norm Adherence Violation Labeling

Given a dialogue and a norm on which the dialogue is based upon, the task has 4 steps:

1. Summarize the Norm in 5 words as Norm

Action

2. Indicate which character in the dialogue performs the Norm Action

3. Repeat every turn and only indicate 'Adhered' when the sentence closely aligns with the Norm. Otherwise, indicate 'Not Relevant'.

4. After each label, provide a short explanation for why the norm is strongly entailed or not relevant at sentence level.

Format:

Repeat each turn in a bracket

Append Adhered or Not Relevant label for each turn

Use "|" to separate role, label and explanation if needed

Norm: In a professional setting with higher status speaking to lower status, it is permitted to use direct language, a strong tone of voice, and display emotions when criticizing one's behavior, ideas, and work.

Dialogue:

张教练: 小陈，进来坐。你今天比赛时的那个失误，不止是你自己比赛历史有了污点，也让我们队失去了比赛胜利的机会。

小陈: 我知道我做错了。

张教练: 而且我强调的不仅仅是你犯的错，而是你没有注意到你思想问题。

张教练: 小陈，你需要更多的多传球给你的队友，不能老是单打独斗。

小陈: 教练我会改正的

张教练: 你今天的投篮还是很不错的，继续努力。

小陈: 谢谢教练。我一定会好好听取你的建议。

张教练: 很好，去休息吧。

Norm Action: offer criticism

Actor of the Norm:

张教练: coach, higher status, criticizer

Dialogue:

(张教练: 小陈，进来坐。你今天比赛时的那个失误，不止是你自己比赛历史有了污点，也让我们队失去了比赛胜利的机会。): Adhered | 张教练 criticizes his player's performance by using direct wordings including "失误", "污点", and "让我们队伍失去"

(小陈: 我知道我做错了。): Not Relevant | 小陈 is not acting the criticism norm

(张教练: 而且我强调的不仅仅是你犯的错，而是你没有注意到你思想问题。): Adhered | 张教练 criticizes 小陈's ideas of how to play basketball by questioning him

(张教练: 小陈，你需要更多的多传球给你的队友，不能老是单打独斗。): Adhered | 张教练 offers a mild criticism by saying "不能老师单打独斗"

(小陈: 教练我会改正的): Not Relevant | 小陈 is not an actor of criticism norm

(张教练: 你今天的投篮还是很不错的，继续努力。): Not Relevant | 张教练 does not criticize here

(小陈: 谢谢教练。我一定会好好听取你的建议。): Not Relevant | 小陈 is not a criticizer

(张教练: 很好，去休息吧。): Not Relevant | not criticism statement

## D   Faithfulness Evaluation of Elaborated Situations

|  | Chinese Situations | US Situations |
|---|---|---|
| Adherence | 98% (57/58) | 90% (45/50) |
| Violation | 100% (56/56) | 74% (40/54) |

Table 4: Faithfulness of the situation elaboration stage for both Chinese and American norm adherences and violations, measured by the percentage of situations that entail the corresponding norms upon which they were generated.

To validate the faithfulness of situation elaborations by ChatGPT, we collect a sample of 218 (*norm, elaborated situation*) pairs. Specifically, for each language (Chinese and English), for each of 10 norm categories, and under norm adherence and violation settings, we randomly select 3 norms along with 2 situations for each norm. Three annotators are then asked to label, for each (*norm, elaborated situation*) pair, if the situation entails the corresponding norm, and to resolve disagreements via majority vote. The final faithfulness score is measured by the percentage of situations that entail the norms in each pair for each language and under norm adherence or violation settings.

Table 4 shows that elaborated situations for Chinese norms achieve high faithfulness scores for both norm adherences and violations. On the other hand, most of the situations elaborated for American norm adherence are also faithful, while those for American norm violation achieve a relatively low faithfulness. In addition, to measure inter-annotator agreements among three annotators, we compute Fleiss Kappa score on the sample of 218 (*norm, elaborated situation*) pairs and individually for norm adherence and violation settings. The

overall Fleiss Kappa score on the entire sample among 3 annotators is 0.51. For 114 norm adherence subsample, Fleiss Kappa score is 0.59; for the remaining 104 norm violation pairs, the Fleiss Kappa score is 0.44.

## E   Topic Models for Generated Situations

To provide greater clarity into the topics present in the situations behind our dialogues, we train a 30-topic LDA model (Blei et al., 2003) and label each topic manually with its most prominent theme. Details are shown in Table 5 and Table 6, below.

| Topic Theme | Top Tokens |
|---|---|
| 0. Compliments | compliments named situation compliment decides insincere feels hair |
| 1. Respects | respect language show chinese formal showing greeting respectful young respected |
| 2. Friend & Social Gathering | friends group friend party situation chinese close social conversation restaurant |
| 3. Gym & Workout | chinese named gym situation china workout american cultural foreigner |
| 4. Name | ming xiao ming's situation david miss named expresses day words |
| 5.Compliment & Humility | compliments chinese work compliment hard responds success humility situation response |
| 6. Religion & Temple | temple service situation master restaurant religious waiter church customer buddhist |
| 7. Apology & Responsibility | jack apology apologize accidentally situation mistake immediately apologizes chinese responsibility |
| 8. Gratitude & Appreciation | chinese situation gratitude china group norm zhou express culture grateful |
| 9. Community & Social Service | event felt situation charity china volunteers community volunteer didn't noticed |
| 10. Family | family parents gathering situation dinner members mother uncle younger father |
| 11. Funeral | family condolences offer deceased situation support chinese members loss show |
| 12. Office | liu colleagues team project meeting work colleague situation chinese company |
| 13. School & Classroom | teacher students student chinese class school professor classroom university classmates |
| 14. Gift Giving | chinese mrs gratitude gift appreciation norm situation show express host |

Table 5: Manually labeled generated situation topics and their top tokens, as captured from a 30-topic LDA model trained on situations generated by ChatGPT.

| Topic Theme | Top Tokens |
| --- | --- |
| 15. Office & Professional Setting | company manager employees senior situation ceo status employee junior |
| 16. Office & Professional Setting | business chinese meeting company potential american situation conference clients card |
| 17. Criticism & Feedback | chinese direct situation request language norm hesitant status approach judge |
| 18. Criticism & Support | feedback criticism situation improve work chinese norm constructive giving |
| 19. Community | group members community situation member approach meeting concerns discussing suggests |
| 20. Name | wei wei's lin named jing store customer norm xia asks |
| 21. Sports & Teamwork | team coach game xiaoming situation players basketball teammates sports performance |
| 22. Taking Leave | leave leaving situation norm social goodbye early chinese attend time |
| 23. Name | zhang zhang's named situation feels lei notices li's decides china |
| 24. Etiquette & Public Transportation | woman man young situation elderly chinese named bus seat susan |
| 25.Government & Officials | liang government status situation official named office yang officials question |
| 26. Music & Concert | audience chinese ling music performance concert situation fans beijing named |
| 27. Wedding | guests wedding bride situation groom ceremony dress reception traditional chinese |
| 28. Argument & Debate | situation chinese behavior argument tom discussion china discussing opinions aggressive |
| 29. Social Events | behavior situation feels uncomfortable norm china starts phone feel social |

Table 6: Manually labeled generated situation topics and their top tokens, as captured from a 30-topic LDA model trained on situations generated by ChatGPT.