# OpenReview forum: "NormDial: A Comparable Bilingual Synthetic Dialog Dataset for Modeling Social Norm Adherence and Violation"
_EMNLP/2023/Conference — EMNLP 2023 Main_

### Official Review · Reviewer_Gicu · 2023-08-03

**Soundness:** 3

**Excitement:**

2: Mediocre: This paper makes marginal contributions (vs non-contemporaneous work), so I would rather not see it in the conference.

**Paper Topic And Main Contributions:**

The authors propose a synthetic dialogue dataset based on adhering/violating social norms. They first augment an existing set of social norms by prompting. Then, they further prompt ChatGPT to generate dialogues based on some pre-determined social norms.

**Questions For The Authors:**

1. What are some use cases for this dataset?
2. How is the diversity of the dataset? You mentioned that diversity is a problem for dialogue generation using prompting. That's why you first prompted ChatGPT for 10 scenarios and then prompt ChatGPT again using these scenarios to generate dialogue. Do you have quantitaive measure of the diversity to justify this approach?

**Reasons To Accept:**

1. Social norm-based dialogue datasets are relatively under-explored.

**Reasons To Reject:**

1. The quality of the model-generated conversations is debatable. The authors claim that the quality of this dataset "exceed the quality of naturally occurring datasets" through human evaluation (Table 2). However, this is questionable because as the authors mentioned, those are expert-annotated datasets, whereas their dataset is generated through prompting.

2. The quality of the model-generated social norms is not analyzed. Although the authors mention that invalid norms are thrown away by the three human annotators, there isn't an analysis of, for example, the inter-annotator agreement for this part.

**Reproducibility:**

2: Would be hard pressed to reproduce the results. The contribution depends on data that are simply not available outside the author's institution or consortium; not enough details are provided.

**Reviewer Confidence:**

3: Pretty sure, but there's a chance I missed something. Although I have a good feel for this area in general, I did not carefully check the paper's details, e.g., the math, experimental design, or novelty.

---

> ### Author Rebuttal · Authors · 2023-08-29
>
> We thank the reviewer for their comments.
>
> **Reason To Reject 1:**
>
> We would like to clarify that the dialogues in both DailyDialogue and LDC are naturally occurring, not expert-annotated. The only part that is expert annotated are social norms from LDC. Hence, our claim that the dialogues generated through prompting exceed quality of naturally occuring datasets is valid, as we show by our detailed human evaluation following strict criteria (see response to Reason to reject 2 and Appendix A). We also note that similar conclusions with respect to synthetic dialogues have been observed in prior work (Chen et. al, 2023).
>
>  Chen, Maximillian, Alexandros Papangelis, Chenyang Tao, Seokhwan Kim, Andy Rosenbaum, Yang Liu, Zhou Yu, and Dilek Hakkani-Tur. 2023. “PLACES: Prompting Language Models for Social Conversation Synthesis.” In Findings of EACL 2023. http://arxiv.org/abs/2302.03269.
>
> **Reason To Reject 2:**
>
> In our work, we mention in line 142-151 that social norms were validated by interpersonal communication experts with knowledge of the culture so they are of high quality.
>
> To ensure both the correctness and detailedness of norm description, our experts used the following detailed criteria to qualitatively evaluate
> * if the model-generated norm aligns with annator’s experience in the society (to make sure the norm is factual)
> * if the norm falls into the desired norm category
> * if the norm describes specific rules/action (instead of being generic)
> * if the norm is specific to Chinese/ American culture (instead of being universal across cultures)
>
> Annotators resolved any disagreements between each other by discussing with each other their labels, so we do not report quantitative IAA metrics.
>
> We will put the details of this criteria that our experts used for evaluating norms in the appendix to address this concern. Our norms are also sufficiently diverse (see lines 120-151), covering 10 distinct categories, and we will also put the results of a 20-topic LDA trained on norms in the appendix D.
>
> **Question 1** What are some use cases for this dataset?
>
> In our Introduction (46 - 52) and Background section (95), we discussed in detail the utility of a LLM- generated dialogue dataset of social norms. We align with prior work in this realm (Chen et al.,  2023; Kim et al., 2022; Kim et al., 2022) but also point a new direction in cross-cultural and cross-lingual dialogues of social norm adherence and violation.
>
> To summarize, our dataset
> 1. Can be explicitly used for model pre-training and fine tuning under supervision to make LLM aware of social norms in different cultures or be able to detect and avoid violation of social norms in different cultures.
>
> 2. Can be used to analyze cross-cultural commonalities and differences in conversations in a variety of life situations for future social science research.
>
> 3. Can be useful for researchers to analyze linguistic characteristics that, for example, might cause norm violation in one language but often observed in another.
>
>
> **Question 2** How is the diversity of the dataset?
>
> To quantify the diversity of the dialogues generated by our CoT prompting method, we randomly selected 5 Chinese and American norms from different norm categories. We compared the distinct n-gram results of dialogues that are generated by going through Scenario Generation, Situation Elaboration, and Dialogue Generation stages versus simply taking the norm to generate dialogues without using situations.
>
> CoT includes dialogues that are generated by going through Scenario Generation, Situation Elaboration, and Dialogue Generation stages, while Simple only takes the norm and directly generates dialogues without using situations. We control the same hyperparameters and the same model (gpt3.5-turbo) for these two prompting methods. The lexical diversity of the dialogues generated by CoT has shown a significant  improvement compared to Simple prompting method. This shows that our approach is effective to improve the diversity of synthetic dialogues and we will include all the details in the Appendix.
>
> More details on the Distinct N-grams results:
>
> 1-gram: 0.24 (CoT)	        0.17 (Simple)
>
> 2-grams: 0.74 (CoT)	0.59 (Simple)
>
> 3-grams: 0.94 (CoT)	0.80 (Simple)
>
> 4-grams: 0.98 (CoT)	0.89 (Simple)
>
>
> **References**
>
> Chen, Maximillian, Alexandros Papangelis, Chenyang Tao, Seokhwan Kim, Andy Rosenbaum, Yang Liu, Zhou Yu, and Dilek Hakkani-Tur. 2023. “PLACES: Prompting Language Models for Social Conversation Synthesis.” In Findings of EACL 2023. http://arxiv.org/abs/2302.03269.
>
> Hyunwoo Kim, Jack Hessel, Liwei Jiang, Peter West, Ximing Lu, Youngjae Yu, Pei Zhou, Ronan Le Bras, Malihe Alikhani, Gunhee Kim, Maarten Sap & Yejin Choi (2022) SODA: Million-scale Dialogue Distillation with Social Commonsense Contextualization. arXiv. https://arxiv.org/abs/2212.10465
>
> Hyunwoo Kim, Youngjae Yu, Liwei Jiang, Ximing Lu, Daniel Khashabi, Gunhee Kim, Yejin Choi & Maarten Sap (2022) ProsocialDialog: A Prosocial Backbone for Conversational Agents. EMNLP. https://arxiv.org/abs/2205.12688

---

### Official Review · Reviewer_q6Mj · 2023-08-05

**Soundness:** 3

**Excitement:**

3: Ambivalent: It has merits (e.g., it reports state-of-the-art results, the idea is nice), but there are key weaknesses (e.g., it describes incremental work), and it can significantly benefit from another round of revision. However, I won't object to accepting it if my co-reviewers champion it.

**Paper Topic And Main Contributions:**

The authors utilized ChatGPT to create dialogues that follow either American or Chinese social norms, with the option to either conform to or defy these norms. Evaluation results indicate that the dialogues generated are more fluent and natural compared to those generated by DailyDialog and the Linguistic Data Consortium.

**Questions For The Authors:**

A. How did the Chain-of-Thought prompting improve the labeling process?

B. How effective was ChatGPT for scenario generation and situation elaboration?

**Reasons To Accept:**

The authors introduced a dialogue dataset that can improve AI's conversational skills by considering social norms. The scenarios presented in the paper could encourage research and facilitate the development of language technology that is culturally aware and sensitive.

**Reasons To Reject:**

The annotations for norm adherence and violations in the generated situations may become misleading as social norms evolve with the progress of society. The author did not present the result to support the claims that Chain-of-Thought prompting improves the labeling process.

**Reproducibility:**

2: Would be hard pressed to reproduce the results. The contribution depends on data that are simply not available outside the author's institution or consortium; not enough details are provided.

**Reviewer Confidence:**

3: Pretty sure, but there's a chance I missed something. Although I have a good feel for this area in general, I did not carefully check the paper's details, e.g., the math, experimental design, or novelty.

---

> ### Author Rebuttal · Authors · 2023-08-29
>
> We thank the reviewer for their comments.
>
> **Reasons to reject:**
> Absolutely, social norms are constantly evolving! This longitudinal aspect must be contended with for any computational social science work around social norms. As our focus is on current social norms, we mitigated the potential possibility of having significantly outdated norms in our data by using a human-AI collaboration framework where experts with significant, contemporary lived experiences in both cultures (10+ years) manually validated the norms for each culture (lines 142-151), before using them to prompt the generation of dialogues that violate or adhere to the given norm. We hope that this framework will allow us and other researchers to continuously generate datasets and capture the dynamic nature of social norms, as they evolve and change over time. We will ensure the release of a data sheet that explains the timeframe of when the data was generated & evaluated, and will add a discussion of this in the limitations section. As a result of this verification, we believe that our bilingual dataset is a valuable resource for current work on cross-cultural norms.
>
> **Question A: How did the Chain-of-Thought prompting improve the labeling process?**
>
> Previous literature shows that Chain of Thought prompting significantly improves reasoning and task performance of language models including classification tasks  (Wei et al., 2022; Kojima et al., 2022, inter alia). Thus, we decided to use CoT to improve our labeling accuracy.
> We have included metrics (precision, recall, F1 score) to evaluate the labeling process in Table 3 of our paper. Moreover, we will also include the results of comparing the labeling accuracy of a sample of 50 dialogues generated with and without the CoT prompting approach in the appendix to address this concern.
>
> The task of labeling dialogue turns with norm adherence/violation is complicated because
>
> a. Norm descriptions tend to be very lengthy. We observed that ChatGPT, without CoT, often fails to focus on the norm action but rather uses information weakly related to the norm itself to label dialogues.
>
> b. Most of our dialogues are multi party. Without using CoT, ChatGPT is often observed to hallucinate turns where a character actually performs the norm versus utterances that serve as context for the norm demonstration, as we discussed in the paper (256 - 264).
>
> We use CoT to address problem a. because by summarizing the lengthy norm description into a short action, ChatGPT is observed to be less prone to use irrelevant information to label dialogues, thus better understanding the task. Similarly, for problem b., because by first identifying characters who perform the norm, ChatGPT is observed to better distinguish utterances where norms actually are adhered/violated versus the context that is not related to the norm.
> Yet, as we mentioned in our paper (260-264), these phenomena remain an exciting field for future research to tackle and address.
>
>
> **Question B: How effective was ChatGPT for scenario generation and situation elaboration?**
>
> We thank the reviewer for this question! On effectiveness, we base our analysis on two main fronts: 1) faithfulness of the situation elaboration against the original situation and 2)  diversity of the generated scenarios (for the scenario generation stage)
>
> Our qualitative analysis showed that ChatGPT was effective across these fronts.
>
> 1. On the faithfulness of the situations elaborated by ChatGPT, our qualitative analysis conducted during human validation shows that the elaborated situations are faithful to the norms as they don’t deviate from the gist of the norm. We will add details of this analysis to the appendix of the final camera-ready version of the paper.
>
> 2. With our Chain-of-Thought prompt and in-context examples, ChatGPT is able to produce diverse scenarios and detailed situations in stage 1 (scenario generation) and 2 (situation elaboration). In Appendix D of our work, we included the results of a 30-topic LDA model trained on situations generated by ChatGPT, which covers a variety of topics from family, office, to funeral.
>
> **References**
>
> Wei, Jason, Xuezhi Wang, Dale Schuurmans, Maarten Bosma, Fei Xia, Ed Chi, Quoc V. Le, and Denny Zhou. "Chain-of-thought prompting elicits reasoning in large language models." Advances in Neural Information Processing Systems 35 (2022): 24824-24837.
>
> Kojima, Takeshi, Shixiang Shane Gu, Machel Reid, Yutaka Matsuo, and Yusuke Iwasawa. "Large language models are zero-shot reasoners." Advances in neural information processing systems 35 (2022): 22199-22213.

---

### Official Review · Reviewer_nowG · 2023-08-07

**Soundness:** 4

**Excitement:**

3: Ambivalent: It has merits (e.g., it reports state-of-the-art results, the idea is nice), but there are key weaknesses (e.g., it describes incremental work), and it can significantly benefit from another round of revision. However, I won't object to accepting it if my co-reviewers champion it.

**Missing References:**

- Chen, Maximillian, Alexandros Papangelis, Chenyang Tao, Seokhwan Kim, Andy Rosenbaum, Yang Liu, Zhou Yu, and Dilek Hakkani-Tur. 2023. “PLACES: Prompting Language Models for Social Conversation Synthesis.” In *Findings of EACL 2023*. http://arxiv.org/abs/2302.03269.
- Hyunwoo Kim, Jack Hessel, Liwei Jiang, Peter West, Ximing Lu, Youngjae Yu, Pei Zhou, Ronan Le Bras, Malihe Alikhani, Gunhee Kim, Maarten Sap & Yejin Choi (2022) **SODA: Million-scale Dialogue Distillation with Social Commonsense Contextualization**. *arXiv*. https://arxiv.org/abs/2212.10465
- Hyunwoo Kim, Youngjae Yu, Liwei Jiang, Ximing Lu, Daniel Khashabi, Gunhee Kim, Yejin Choi & Maarten Sap (2022) **ProsocialDialog: A Prosocial Backbone for Conversational Agents**. *EMNLP*. https://arxiv.org/abs/2205.12688

**Paper Topic And Main Contributions:**

This short paper presents NormDial, a new dataset of English and Chinese dialogues annotated with relevant social norms along with labels on whether the norm is being followed or violated. Authors use a machine generation with human verification pipeline to create the dataset, which contains 4k dialogues and 30k conversational utterances: authors first generate a set of social norms with ChatGPT, then flesh out the social norms into dyadic social situations, from which a dialogue is then generated. The English and Chinese conversations are then verified for naturalness, interestingness, and coherence, as well as whether the speakers follow the norms of the respective cultures. Finally, authors perform a small investigation of whether ChatGPT can accurately predict whether a conversation follows/violates a social norm, finding that it struggles to detect norm-violating utterances.

**Reasons To Accept:**

- I appreciate the fact that the dataset is in a language other than English, and that a non-Western culture is being included.
- The machine-and-human creation of the dataset is reasonable and effective, and I particularly appreciated the careful manual verification of the dialogues.
- The experiments on ChatGPT (§4) are well-executed and yield some interesting insights.

**Reasons To Reject:**

This work is sound and reasonable. However, the paper missed some important related work that was not covered, making some claims of novelty not well supported. Specifically, Kim et al. (2022a) introduced the ProSocial Dialogues corpus which contains English conversations with social norms annotated at the utterance level. Additionally, the machine-generated-human-verified pipeline to generate social situations which are fleshed out into dialogues was introduced by Kim et al (2022b) and Chen et al. (2023). I suggest the authors edit their introduction to better scope their claims of novelty, and include these works into the related work section. That being said, the paper's main novelty lies in the cross-cultural nature of the dataset, which is a huge asset for the NLP community!

**Reproducibility:**

2: Would be hard pressed to reproduce the results. The contribution depends on data that are simply not available outside the author's institution or consortium; not enough details are provided.

**Reviewer Confidence:**

4: Quite sure. I tried to check the important points carefully. It's unlikely, though conceivable, that I missed something that should affect my ratings.

---

> ### Author Rebuttal · Authors · 2023-08-29
>
> We thank the reviewer for their comments and recognizing novelty of our work! We also note that we will be releasing all the data and codebase to encourage reproducibility and further research.
>
> * Reason to Reject 1:
> We thank the reviewers for pointing out some papers that are very relevant to ours. To clarify on the first related work listed as missing (Chen et al., 2023), we cite this paper multiple times throughout our paper, for example on the Introduction (49) and Background and Related Work (76-77) sections. We will emphasize more on the cross-cultural and bilingual nature to highlight the novelty and contributions of our work, as the reviewer correctly points out! And as the reviewer advised, we will include the other two papers in the background section.

---

### Meta-Review · Area_Chair_a1EZ · 2023-09-12

**Recommendation:** 4

**Metareview:**

This short paper introduces a dataset of dialogues which are annotated as to whether they adhere to social norms. The dataset contains English and Chinese dialogues. The dialogues are synthetic - they were generated using ChatGPT and then annotated. The authors also evaluate whether ChatGPT can recognize norm violations and find that it does not do very well on that task.
The reviewers appreciate the cross-cultural social norms challenge that this dataset can help to address.
The reviewers also point out some shortcomings with respect to missing citations, unsupported claims on CoT reasoning effects and unsupported claims regarding the quality of the proposed dataset. However, the authors convincingly address these points in their rebuttal, which should flow into a final version of this paper.

---

### Decision · Program_Chairs · 2023-10-07

**Decision:**

Accept-Main

**Comment:**

This short paper introduces a dataset of dialogues which are annotated as to whether they adhere to social norms. The dataset contains English and Chinese dialogues. The dialogues are synthetic - they were generated using ChatGPT and then annotated. The authors also evaluate whether ChatGPT can recognize norm violations and find that it does not do very well on that task.
The reviewers appreciate the cross-cultural social norms challenge that this dataset can help to address.
The reviewers also point out some shortcomings with respect to missing citations, unsupported claims on CoT reasoning effects and unsupported claims regarding the quality of the proposed dataset. However, the authors convincingly address these points in their rebuttal, which should flow into a final version of this paper.